# Trends in Malignant and Benign Brain Tumor Incidence and Mobile Phone Use in the U.S. (2000–2021): A SEER-Based Study

**DOI:** 10.3390/ijerph22060933

**Published:** 2025-06-13

**Authors:** Li Zhang, Joshua E. Muscat

**Affiliations:** 1Department of Public Health Sciences, Penn State College of Medicine, 500 University Drive, Hershey, PA 17033, USA; lzhang9@pennstatehealth.psu.edu; 2Center for Research on Tobacco and Health, Penn State Cancer Institute, Pennsylvania State University, Hershey, PA 17033, USA

**Keywords:** brain cancer, incidence, time trends, mobile phone, SEER

## Abstract

(1) Background: There has been an ongoing concern for several decades that radiofrequencies emitted from mobile phones are related to brain cancer risk. We calculated temporal trends in brain cancer incidence rates in adults and children and compared them to mobile phone subscription data over the same time period. (2) Methods: We analyzed the Surveillance, Epidemiology and End Results (SEER 22) cancer database between 2000 and 2021. Age-standardized incidence rates (ASR) per 100,000 people were calculated and the annual percentage change (APC) for malignant and benign brain cancer and vestibular schwannomas (acoustic neuromas of the 8th cranial nerve) was established. The total number of mobile phone subscriptions in the United States was plotted for the period 1985–2024. (3) Results: The APC for adolescents and adults was −0.6 (*p* = 0.0004) for malignant tumors, −0.06 (*p* = 0.551) for temporal lobe tumors, and 1.9 (*p* = 0.00003) for benign tumors. The APC for benign acoustic neuroma was 0.09 (*p* = 0.8237), suggesting that mobile phone use is unlikely to be associated with this tumor type. There was a 1200-fold increase in the number of cell phone subscriptions during this period. (4) Conclusions: These findings suggest that mobile phone use does not appear to be associated with an increased risk of brain cancer, either malignant or benign.

## 1. Introduction

After the market introduction of mobile phones in 1984, there has been an ongoing concern about their possible health effects, including the risk of brain cancer [1,2,3,4,5]. Numerous organizations have evaluated the scientific literature [6,7,8,9,10,11,12]. The National Cancer Institute and the Food and Drug administration’s position is that cell phone use does not cause cancer of any kind [6,7].

In contrast, radiofrequency waves from mobile phones, a form of nonionizing radiation, were classified as possibly carcinogenic to humans (Group 2B) by the International Agency for Research on Cancer [13]. The evidence was considered as being limited for glioma and acoustic neuroma and inadequate for other types of cancers. The literature supporting these evaluations were primarily case–control in design. While increased risks were found in some studies, study biases might have contributed to these associations [13,14,15,16,17]. In particular, exposure characterization and selection bias are considered the types of biases that are most likely to have affected the findings. In addition, unique to this literature is the reported side of the head where the phone was used. More recent studies that were able to measure the effects of longer-term mobile phone use (10–15 years) have mostly found no increased risks [18,19,20,21,22,23,24,25,26,27,28,29,30,31].

One of the methodologies used in these evaluations is an examination of cancer trends in well-established population-based cancer registries. The number of mobile phone subscribers, including children and adolescents, has exponentially increased since the end of 1999, and it might be expected that this would affect the rates of brain cancer if an association exists. An analysis of incidence rates in the U.K. and Scandinavian countries indicated that any trends were incompatible with hypothetical levels of increased risk [24,32]. Similarly, in New Zealand and in the Central Brain Tumor Registry of the United States, the stable incidence of brain cancer did not indicate a risk regarding mobile phones in relation to total brain cancer or when further analyzed by specific lobes in the cortex or histologic type [33,34]. These earlier studies tracked patterns up until about 2010, but questions of potential long-term latency remain. To fill this gap, in the current study, we analyzed long-term incidence data trends from 2000 to 2021 in the population-based Surveillance, Epidemiology, and End Results Program (SEER), and compared these with commercial data on mobile phone subscription usage over the same period. We examined the incidence trends of malignant tumors, including temporal lobe tumors, benign brain cancer, and benign acoustic neuroma (cancer of the eighth cranial nerve, also called vestibular schwannoma). Almost all studies of cell phones and cancer have been case–control designs, and to a lesser extent cohort studies. Another methodology that can be used to address this question is the analysis of trends, where rising incidence or mortality indicates a potentially new risk factor. This might be especially useful to account for a long hypothetical latency. One reason why an analysis of trends adds information to the literature is that earlier studies that showed no effects could not rule out a possible effect due to possible long latency periods. There is no a priori basis for assuming what the length of possible latency periods should be. The latency period of radiation-induced solid tumors is at least 10 years, and so much of the literature has not accounted for potentially long-term latency periods.

## 2. Materials and Methods

### 2.1. Data Sources and Acquisition

SEER is a network of population-based cancer registries in geographic areas that represent the racial and ethnic populations in the US [35]. SEER 22 is the most recent version of the SEER database, which covers the years 2000–2021 in 22 US central cancer registries, representing 47.9% of the US population [36].

The current analysis examined the time trends of incidence rates of overall benign (behavior classified as borderline and in situ) and malignant (behavior as malignant) brain cancer (including ICD-O-3 site and histology codes: brain—C710–719 (excluding 9050–9055, 9140, 9530–9539, 9590–9993); cranial nerves other nervous system—C710-C719 (9530–9539), C700–C709, and C720–C729 (excluding 9050–9055, 9590–9993)) [37]. In addition, we conducted a separate analysis for brain regions that receive the highest dose of radiofrequency exposure from cell phones. Tumors that occur in these areas include temporal lobe cancers (C71.2) and acoustic neuromas from the eighth cranial nerve (C72.4). A sub-analysis was also conducted focusing on children and adolescents aged <20 y for overall malignant brain tumors and malignant temporal lobe tumors. Metastatic tumors from other sites were not included.

The total number of mobile phone subscriptions during the period from 1985 to 2021 was sourced from the US Bureau of Economic Analysis using open World Bank data [38].

SEER*Stat 8.4.3 was used to measure annual percentage changes in cancer rates and significant trends in rates [39]. We determined incidence rates for individuals aged 10 years and over (likely to be mobile phone users) who received a diagnosis of brain cancer from 1 January 2000 to 31 December 2021. Survey data indicate that the age at first use is about 12.2 y ± 2.0, although it might have been older in previous decades [40]. The year 2000 was chosen based on the hypothetical latent effect and consistency with the SEER 22 study period. The latency period of radiation-induced solid tumors is at least 10 years [41]. Since SEER 22 did not include benign brain cancers until 2004, benign brain cancer incidence rates were determined for the period 2004–2021.

### 2.2. Statistical Analysis

All brain cancer incidence rates, reported per 100,000 persons, were calculated using SEER*Stat, version 8.4.3, and age-adjusted to the 2000 US standard population (19 age groups, P25-1130) [42]. Assuming that the linearity in regression trends holds, by fitting linear regression models with each calendar year at diagnosis (2000–2021) as an independent variable, changes (significant increasing/decreasing or stable) in time trends of age-adjusted incidence rates were assessed using the annual percentage change (APC) based on least square methods and corresponding intervals (CIs) estimated by SEER*Stat 8.4.3, as recommended by the SEER website [39,42]. Statistically significant APCs were different from zero based on a two-sided *p* < 0.05.

### 2.3. Ethical Approval

This study was exempt from Penn State’s IRB review, since the data is de-identified and publicly available from SEER.

## 3. Results

### 3.1. Study Population Characteristics

From 2000 to 2021, this study included 200,568 children and adults (aged ≥ 10) diagnosed with malignant brain cancer. The study also included 375,895 children and adults with a benign brain cancer diagnosis (2004 to 2021), including 40,870 benign acoustic neuromas.

### 3.2. Mobile Phone Subscriptions and Time Trends of Brain Cancer Incidence Rates

The number of mobile phone subscriptions in 1985 was 0.3 million, increasing to 109.5 million in 2000 and 361.7 million in 2021 (Figure 1). This represents a 1200-fold increase over this period.

During the study period 2000–2021, the incidence rates of malignant brain cancers for adults and children decreased (APC = −0.62, 95% CI: −0.73, −0.51; *p* = 0.0004, Figure 2). There was no change in the incidence rates of malignant temporal lobe tumors (APC = −0.06, 95% CI: −0.28, 0.16; *p* = 0.551). Between 2004 and 2021, the incidence rates of overall benign brain cancers increased (APC = 1.9, 95% CI: 1.33, 2.48; *p* = 0.00003, Figure 3). The rates of acoustic neuroma remained stable (APC = 0.09, 95% CI: −0.71, 0.88; *p* = 0.8237). The lack of significant change in the incidence of benign acoustic neuroma (*p* = 0.8237) suggests that mobile phone use is unlikely to be associated with this tumor type. For children and adolescents, malignant temporal lobe incidence rate decreased significantly (APC = −0.99, 95% CI: −1.87, −0.11; *p* = 0.0297, Figure 4) between 2000 and 2021; however, there was no change in malignant brain tumor rates in this age group (APC = −0.23, 95% CI: −0.63, 0.18; *p* = 0.2537).

## 4. Discussion

Mobile phones were introduced to the market around 1985. Within two and a half decades, the number of subscriptions in the U.S. increased to 361 million by 2021, surpassing the current US population [43]. During this period, the rates of most forms of malignant and benign brain cancer remained stable for both adults and adolescents. This included the rates of brain cancer in the temporal lobe and the eighth cranial nerve, which are the intracranial areas with the highest degree of exposure to radiofrequency radiation in mobile phone users. The incidence rates for benign tumors (primarily meningiomas) were also stable, but did increase slightly over this period [44]. This is unlikely to be due to mobile phone use. Since 2000, there has been an annual increase in the use of noninvasive medical imaging for accurately diagnosing and treating diseases/conditions of the head, which increased the incidental detection of asymptomatic meningiomas [45,46]. The possibility that mobile phone use is weakly associated with these tumors cannot be completely ruled out by the current data. However, the location of most meningiomas is in cranial areas that have less than the peak specific absorption rate (SAR) [10]. SAR measures the rate at which radiofrequency radiation is absorbed from a cell phone using models that simulate head exposure. The maximum value allowed is set by the Federal Communications Commission. Meningioma of the bone covering the cranial nerves in the inner ear are rare. In a meta-analysis of eight studies, the risk for meningioma was associated with mobile phones [odds ratio (OR) = 0.90; 95% confidence interval (CI) 0.83–0.99] [47].

A simple method to evaluate the possible effects of mobile phones on brain cancer used in this study is comparing the population rates of mobile phone subscriptions with the incidence rate. Inskip et al. conducted a similar analysis for the years 1985–2009 and concluded that mobile phones were not related to the rates of cancer [29]. One recent study is similar to ours. In Taiwan over a 20-year period, there was a weak positive trend in malignant brain cancer rates corresponding with increasing mobile phone use. Although the authors state the findings are inconclusive, the method supports the use of population-based cancer registries to address the issue of mobile phone safety [48]. A systematic review of 63 studies of radiofrequency fields on cancer risk commissioned by the World Health Organization concluded with moderate certainty a lack of risk from mobile phone use [49]. Inherent in these studies are the usual concerns in study designs: exposure assessment validity, recall bias, and latency issues. Among these, the recent Cohort Study on Mobile Phones and Health (COSMOS) is a multi-center study in Europe that had detailed self-reported phone usage as well as supplemental baseline data on phone usage from the network operator [22]. This study did not find associations of lifetime mobile phone use with glioma, acoustic neuroma, or meningioma. In the UK’s Million Women Study, there was also no association of mobile phone use with temporal lobe gliomas [24].

The association of brain cancer with mobile phone use is less well-documented in children than in adults. The largest study is a multicenter case–control study conducted in Denmark, Sweden, Norway, and Switzerland for children and adolescents aged 7–19 years who were diagnosed with a brain tumor between 2004 and 2008 [50]. There was no association with regular use or any dose–response relationship or risk in the temporal lobe. In a subgroup of 24 case and 25 control participants with about 3 years of use as documented by mobile phone company data, the risk was increased (OR = 2.15, 95% CI 1.07–4.29). A concern in children is that their exposure may exceed that of adults due to their thinner cranial bones. Some SAR simulation studies, however, have shown similar exposures for children and adult phone users [51]. It is widely established that there is no known mechanism by which radiofrequency fields can cause cancer (https://www.cancer.gov/about-cancer/causes-prevention/risk/radiation/electromagnetic-fields-fact-sheet, accessed on 28 May 2025). It does not damage DNA. Studies have attempted to find possible indirect mechanisms, such as changes in melatonin levels or oxidative stress, but there is little convincing evidence. Most animal studies have not shown any relationship with tumor incidence.

The current study was based on the use of a population-based incidence surveillance database and commercial data on phone subscriptions. The comparison of these two datasets allows for a simple and inexpensive method to assess the concerns about phone use and brain cancer. The findings indicate that the incidence rates have remained stable, whether mobile phones are used by nearly the entire population or only a small percentage. The analysis also allowed for the possibility of any latent effect that might not be detected in shorter-term case–control and cohort analyses. The lack of an increase in temporal lobe or acoustic neuroma tumors—the areas most exposed to RF radiation—suggests that mobile phones may not contribute to the brain cancer.

There is no known biological mechanism by which cell phone radiofrequency exposure causes brain tumors. Due to their widespread use, especially their quick adaptation by the majority of the population, the precautionary principle dictates the need to conduct such studies. For example, cell phone research has uncovered effects on physiological systems such as brain glucose metabolism [10]. It is assumed that any possible risk from cell phones is highest in the brain topographic areas that have relatively high SARs, within the maximum levels approved by the Federal Communications Commission. Peak rates generally occur behind the ear where the phone is placed, including the topical areas of the temporal lobe, The SAR values depend on phone type, the antenna position, and head shape [52]. SAR measurement methods using phantom head models vary by experimental condition, but the SARs for most adults and children decrease as the RF travels through tissue [53,54]. It may be that the best test of mobile phone safety and brain cancer is an examination of rates in the temporal lobe of the brain and for acoustic neuromas.

There are several limitations to the current study. The study design is ecologic, which limits causal inference. While the study assesses population-level trends, individual exposure levels are not captured. The proxy measure (mobile phone subscriptions) does not account for frequency, duration, or type of use, which affect total exposure to the head. Future studies should incorporate more commercial granular usage data at the population level. It might be expected that the frequency of usage has also increased over this period, although this may be due to the use of smartphone functions such as texting and the use of mobile apps that do not involve placement of the phone against the head. The current study design is ecologic in nature and does not calculate risk ratios and risk ratios for ipsilateral phone use. It should be noted that SEER excluded 2020 data from estimation from joinpoint trends because of delayed reporting due to the COVID-19 pandemic. Also, for benign brain tumors, the data were restricted to the start year of 2004 due to reporting issues. We did not conduct joinpoint analysis and thus decided to include the year 2020. While SEER is a national registry, it covers about 30% of the U.S. population. This may limit its generalizability to the larger US population to some degree. There were some slight revisions in ICD-O codes for meningioma classification in 2016 [55]. However, this would not affect the reported incidence of these tumors. Also, potential confounding variables such as changes in environmental exposures or diagnostic practices over time could not be adjusted for in the models due to a lack of this information in SEER. Finally, cell phone technology has changed substantially over this period, especially with the widespread adoption of smartphones in 2007. However, the SAR limits have not changed for several decades.

## 5. Conclusions

The most reasonable interpretation of our findings is that the use of mobile phones is not a factor in brain cancer development, including for brain areas that experience peak SARs. This conclusion is based on the large increase in cell phone usage over the study period but stable rates of brain cancer. Longitudinal cohort studies with operator-recorded usage data or advanced imaging biomarkers for early detection are needed in the near future to confirm these findings.

## Figures and Tables

**Figure 1 ijerph-22-00933-f001:**
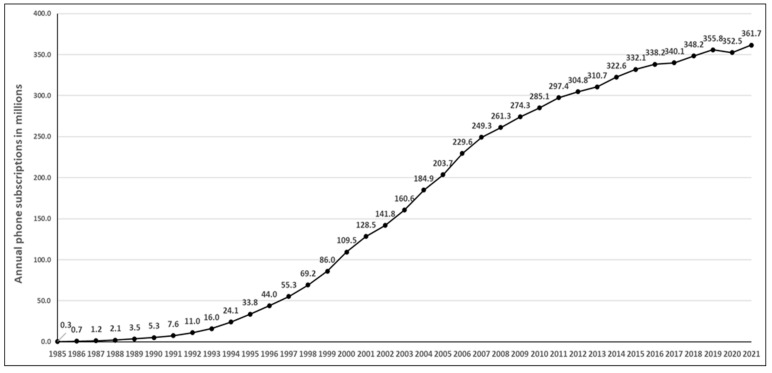
Mobile phone subscriptions in the United States (1985–2021).

**Figure 2 ijerph-22-00933-f002:**
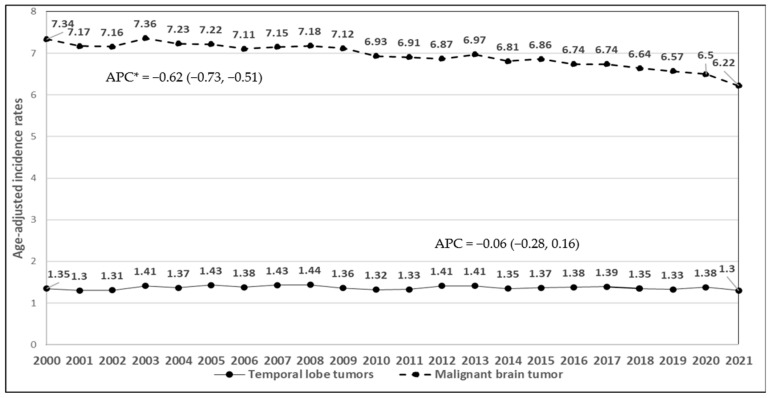
Time trends of age-adjusted incidence rates of malignant and malignant temporal lobe tumors in individuals aged 10 years and older, 2000–2021 (APC*: APC is statistically significant at the 0.05 level).

**Figure 3 ijerph-22-00933-f003:**
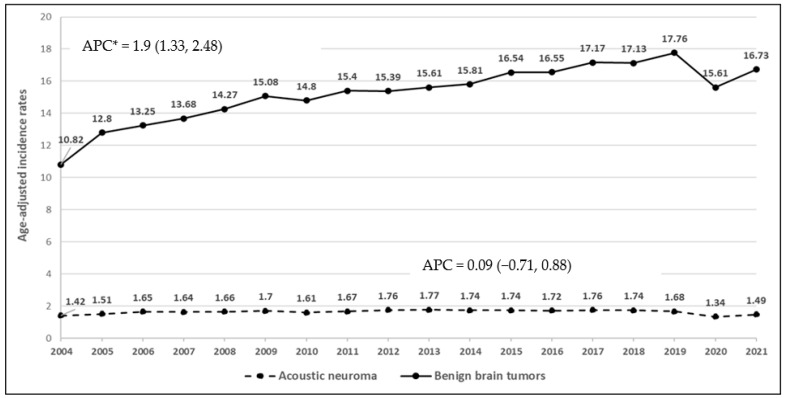
Time trends of age-adjusted incidence rates of benign brain tumors and acoustic neuroma in individuals aged 10 years and older, 2004–2021 (APC*: APC is statistically significant at the 0.05 level).

**Figure 4 ijerph-22-00933-f004:**
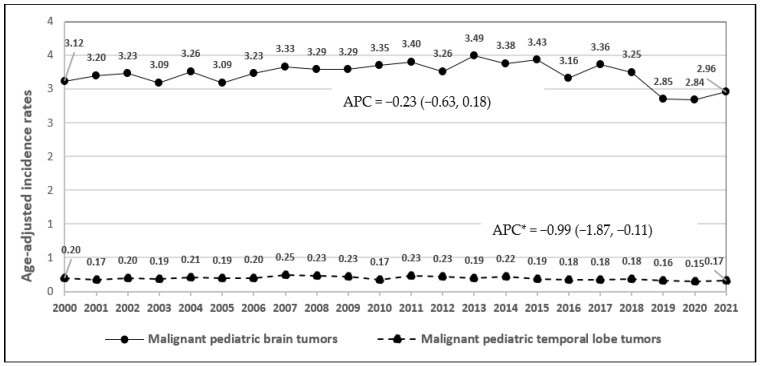
Time trends of age-adjusted incidence rates of malignant and malignant temporal lobe tumors in children and adolescents, 2000–2021 (APC*: APC is statistically significant at the 0.05 level).

## Data Availability

Data is available on the SEER website, and the analytic methods will be made available pending email requests to the corresponding author. Coding algorithms and supplemental information are available pending e-mail request to the corresponding author.

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
