# Peer review of "Trends in Malignant and Benign Brain Tumor Incidence and Mobile Phone Use in the U.S. (2000–2021): A SEER-Based Study"

_ijerph, 2025, doi:10.3390/ijerph22060933_

Round 1

Reviewer 1 Report

Comments and Suggestions for Authors

The study, despite being purely ecological, does provide some additional data on long-term exposure and latency of cell phone use and rates of brain cancer. The paper is straight-forward and well-written. Figure 2 presents malignant tumors in children and adolescents but rates in  adults are not shown. The text does not specify that the APC are for pediatric cancers. Figure 3 combines all ages. It would be informative to show rates in children/adolescents for both malignant and benign, and separately for adults for malignant and benign. Currently, there is no information on malignant brain cancer rates in adults. 

Minor comment:

For Figure 3, the APC is not included in the figure (it is for Figure 2). 

Author Response

The study, despite being purely ecological, does provide some additional data on long-term exposure and latency of cell phone use and rates of brain cancer. The paper is straight-forward and well-written. Figure 2 presents malignant tumors in children and adolescents but rates in  adults are not shown. The text does not specify that the APC are for pediatric cancers. Figure 3 combines all ages. It would be informative to show rates in children/adolescents for both malignant and benign, and separately for adults for malignant and benign. Currently, there is no information on malignant brain cancer rates in adults.

Answer: We thank reviewer for the comments.  We revised Figure 2 which will only reflect the APCs for the malignant brain tumors and malignant temporal lobe tumors for adults and children together.  Also, we will include the APCs for malignant brain tumors and temporal lobe tumors for children and adolescents in the Figure 4 (The rates of other brain cancers are rare in children, and the statistical power for any sensitivity analysis is likely low). 

Minor comment:

For Figure 3, the APC is not included in the figure (it is for Figure 2).

Answer: We thank the reviewer for the comments, and we made revisions.

Reviewer 2 Report

Comments and Suggestions for Authors

The manuscript could not be accepted in the current form for the following reasons:

There are mismatches between the citations mentioned in the text and the references listed in the manuscript.

Introduction:

  1. In introduction, authors mentioned that some case-control studies have suggested increased risk but also potential biases. Authors are requested to briefly specify what these biases are which can strengthen the rationale for using registry-based trend analysis.
  2. Why malignant and benign tumor and specific subtypes like acoustic neuroma, the authors are requested to provide some brief justification for these tumor subtypes.
  3. It is unclear in the manuscript that what does SAR indicate?

    Methodology:

    1. The benign brain tumors data analysis begins in 2004 based on the availability of SEER data. Effects of the analysis should be discussed.
    2. The methods do not address adjustment for potential targets (e.g., changes in technology, or other environmental exposures) that could influence incidence rates over time. The authors are requested to discuss how these factors might affect trend interpretation.
    1. The rationale for the selected study period (2000–2021) is understandable, but a brief explanation on how this window relates to hypothesized latency for brain tumor from mobile phone exposure would strengthen the hypothesis.
    2. The use of age over 10 as the lower limit for the analysis is justified by the current age of first use, but sensitivity analysis using different age cut-offs may be investigated, given varying trends in take-up of mobile phones among youth.

    Discussion:

    1. The authors are requested to rewrite the discussion briefly in correlating to such research as those that have reported weak or borderline associations—such as the Taiwan study reporting a 0.5% increase in brain cancer incidence among smartphone users, before going on to describe how such findings are contrasted with the far larger body of research showing no significant associations.

    Conclusion:

    1. The authors need to recheck and highlight all the abbreviations at the time of first use in the manuscript.

    2. Authors are requested to rewrite the conclusion by including their findings or observations

Author Response

Introduction

In the introduction, the authors mentioned that some case-control studies have suggested increased risk but also potential biases. Authors are requested to briefly specify what these biases are, which can strengthen the rationale for using registry-based trend analysis.

Answers.  We thank the reviewer for the comments. We revised the introduction accordingly in lines 38-41.

While increased risks were found in some studies, study biases might have contributed to these found associations [13-17]. In particular, exposure characterization and selection bias are considered the types of biases that are most likely to have affected the findings. In addition, unique to this literature, is the reported side of the head where the phone was used.

Why malignant and benign tumor and specific subtypes like acoustic neuroma, the authors are requested to provide some brief justification for these tumor subtypes.

Answers.  We thank the reviewer for the comments. Lines 76-80 provide the rationale for the focus on these subtypes.

It is unclear in the manuscript that what does SAR indicate?

Answers.  We have added to the end of the first paragraph in the discussion (lines 162-165) that SAR (Specific Absorption rate) measures the rate at which radiofrequency radiation is absorbed from a cell phone using models that simulate head exposure. The maximum value allowed is set by the Federal Communications Commission.

Methodology:

The benign brain tumors data analysis begins in 2004 based on the availability of SEER data. Effects of the analysis should be discussed.

Answers.  We thank the reviewer for the comments.  We made revision accordingly as follows in the discussion (lines 236-237).

Also, for benign brain tumors, the data were restricted to the start year of 2004 due to reporting issues, which will not reflect the previous trends.

The methods do not address adjustment for potential targets (e.g., changes in technology, or other environmental exposures) that could influence incidence rates over time. The authors are requested to discuss how these factors might affect trend interpretation.

Answers: We thank the reviewer for the comments. We have added a sentence at the end of the limitations section (lines 244-246) that Cell phone technology has changed substantially over this period, especially with the widespread adoption of smart phones beginning in 2007. However, the SAR limits have not changed for several decades.

The rationale for the selected study period (2000–2021) is understandable, but a brief explanation on how this window relates to hypothesized latency for brain tumor from mobile phone exposure would strengthen the hypothesis.

Answers: We added to the end of the Introduction (lines 64-69) that One reason why an analysis of trends adds information to the literature is that earlier studies that showed no effects could not rule out a possible effect due to possible long latency periods. There is no a priori basis for assuming what the length of possible latency periods should be. The latency period of radiation-induced solid tumors is at least 10 years, so much of the literature has not accounted for potentially long-term latency periods.

The use of age over 10 as the lower limit for the analysis is justified by the current age of first use, but sensitivity analysis using different age cut-offs may be investigated, given varying trends in take-up of mobile phones among youth.

Answers: We appreciate this comment. However we decided not to perform this analysis. The rates of these cancers are rare in children, and the statistical power for any sensitivity analysis is likely low.

The authors are requested to rewrite the discussion briefly in correlating to such research as those that have reported weak or borderline associations—such as the Taiwan study reporting a 0.5% increase in brain cancer incidence among smartphone users, before going on to describe how such findings are contrasted with the far larger body of research showing no significant associations.

Answers: We thank the reviewer for the comments. We made revision accordingly in the discussion (lines 172-176):

One recent study is similar to ours. In Taiwan over a 20-year period, there was a weak positive trend in malignant brain cancer rates corresponding with increasing mobile phone use.  Although the authors state the findings are inconclusive, the method supports the use of population-based cancer registries to address the issue of mobile phone safety.

The authors need to recheck and highlight all the abbreviations at the time of first use in the manuscript.

Answers: We thank the reviewer for the comments. We made revision accordingly.

Authors are requested to rewrite the conclusion by including their findings or observations

Answers: We thank the reviewer for the comments. Conclusion has been rewritten as such in lines 249-252:

The most reasonable interpretation of our findings is that the use of mobile phones is not a factor in brain cancer development, or for brain areas that experience peak SARs. This is based on the exponential increase in cell phone usage over the study time period but stable rates of brain cancer.

Reviewer 3 Report

Comments and Suggestions for Authors

The manuscript presents a timely and relevant analysis of the long-term trends in brain cancer incidence in relation to mobile phone usage. The use of SEER 22 data and national mobile phone subscription statistics provides a valuable epidemiological perspective. However, to strengthen the scientific rigor, the following points should be addressed:

  1. The current title does not clearly reflect the scope of the study. To better capture the manuscript’s focus on both malignant and benign tumors, consider revising the title to: "Trends in Malignant and Benign Brain Tumor Incidence and Mobile Phone Use in the U.S. (2000–2021): A SEER-Based Study."

  1. The abstract should more explicitly state the main conclusion. A more direct phrasing would clarify the findings: “These findings suggest that mobile phone use does not appear to be associated with an increased risk of brain cancer, either malignant or benign.”

  1. While international studies (e.g., from the U.K. and Scandinavia) are mentioned in the introduction, the comparison with U.S. data could be made more explicit. A brief synthesis of how the SEER data findings align with these earlier studies would strengthen the rationale.

  1. The methodology section would benefit from more detail regarding how tumors were categorized (e.g., ICD-O-3 codes for malignant and benign tumors) and which demographic variables were included. Clarifying this would improve reproducibility and transparency.

  1. The results mention that the p-value for benign acoustic neuroma (p = 0.8237) was not significant, but the implication of this result could be better explained. Consider stating clearly: “The lack of significant change in the incidence of benign acoustic neuroma (p = 0.8237) suggests that mobile phone use is unlikely to be associated with this tumor type.”

  1. The manuscript contains several minor grammatical and typographical issues. For example, the phrase “that that” in the introduction (line 31) should be corrected. A thorough language and grammar check throughout the document is recommended.

  1. The introduction should better articulate the rationale for this study by highlighting the gap it addresses—particularly the need for up-to-date, U.S.-based trend data given rising mobile phone usage. The ecological design and absence of individual-level exposure data should also be acknowledged upfront.

  1. The discussion section should be expanded to further engage with the broader literature, including large-scale cohort studies and meta-analyses. This will place the current findings in better context and demonstrate their relevance within the wider field.

  1. While the statistical analysis is sound, the manuscript should acknowledge potential confounding variables such as changes in environmental exposures or diagnostic practices over time. This discussion will enhance the validity of the interpretations.

  1. The graphs and tables are generally well-constructed but could be improved with clearer axis labels, larger font sizes, and more descriptive captions to aid reader comprehension. If possible, visual indicators of statistical significance should be added to key graphs.

  1. The observed increase in benign tumors—particularly meningiomas—warrants a more detailed explanation. The authors suggest improved imaging is responsible; however, citing evidence for this trend (e.g., increase in MRI usage) would strengthen this interpretation.

  1. The study design, being ecological, limits causal inference. This should be emphasized more clearly in both the discussion and the abstract. A brief acknowledgment of this limitation in the conclusion would also enhance transparency.

  1. While the study assesses population-level trends, individual exposure levels are not captured. The proxy measure (mobile phone subscriptions) does not account for frequency, duration, or type of use, which are essential factors. The authors should emphasize this limitation and suggest the need for future studies incorporating more granular usage data.

  1. The discussion on biological mechanisms is underdeveloped. The authors should include either relevant experimental evidence or propose potential mechanisms through which RF-EMF exposure might theoretically affect brain tissue. Alternatively, explain more clearly why no mechanism has been established despite widespread exposure.

  1. Given the importance of biological plausibility in interpreting epidemiologic trends, the manuscript would be significantly improved by either conducting or referencing experimental studies. This could include: In vitro or in vivo studies examining RF-EMF effects on neural tissues. SAR distribution simulations in adult and pediatric head models. Studies investigating oxidative stress, DNA damage, or apoptosis due to RF exposure. If direct experiments are not possible, referencing high-quality experimental literature would help bridge the gap between epidemiological and biological evidence.

  1. The issue of latency should be discussed more extensively. Although the 20-year study window is substantial, it remains possible that some cancers, particularly in younger users, might manifest beyond this period. More detailed age-stratified analyses would be useful.

  1. The authors should provide clearer interpretations of non-significant findings. For example, the lack of increase in temporal lobe or acoustic neuroma tumors—areas most exposed to RF radiation—should be emphasized as important negative evidence.

  1. Statistical model assumptions (e.g., linearity in regression trends) should be briefly described in the methods. Clarifying why SEER*Stat was appropriate and what assumptions were made would improve methodological transparency.

  1. The conclusions should include more forward-looking suggestions. For example, recommend longitudinal cohort studies with operator-recorded usage data or advanced imaging biomarkers for early detection.

Author Response

The current title does not clearly reflect the scope of the study. To better capture the manuscript’s focus on both malignant and benign tumors, consider revising the title to: "Trends in Malignant and Benign Brain Tumor Incidence and Mobile Phone Use in the U.S. (2000–2021): A SEER-Based Study."

Answer: We thank the reviewer for the comments. We have adopted this title for our study.

The abstract should more explicitly state the main conclusion. A more direct phrasing would clarify the findings: “These findings suggest that mobile phone use does not appear to be associated with an increased risk of brain cancer, either malignant or benign.”

Answer: We thank the reviewer for the comments. We have adopted the recommended revision in the abstract of the study.

While international studies (e.g., from the U.K. and Scandinavia) are mentioned in the introduction, the comparison with U.S. data could be made more explicit. A brief synthesis of how the SEER data findings align with these earlier studies would strengthen the rationale.

Answer: We thank the reviewer for the comments.  We have revised this paragraph accordingly as follows (Lines 48-60):

An analysis of incidence rates in the U.K. and Scandinavian countries indicated that any trends were incompatible with hypothetical levels of increased risk [24, 32]. Similarly, in New Zealand and in the Central Brain Tumor Registry of the United States, the stable incidence of brain cancer did not indicate a risk for mobile phones with total brain cancer or when further analyzed by specific lobes in the cortex or histologic type [33, 34]. These earlier studies tracked patterns up until about 2010, but questions of potential long-term latency remain. To fill up this gap, in the current study, we analyzed long-term incidence data trends from 2000-2021 in the population-based Surveillance, Epidemiology, and End Results Program (SEER), and compared that with commercial data on mobile phone subscription usage over the same period. We examined the incidence trends of malignant tumors including temporal lobe tumors, benign brain cancer and trends in the incidence rates of benign acoustic neuroma (cancer of the eighth cranial nerve, also called vestibular schwannoma).

The methodology section would benefit from more detail regarding how tumors were categorized (e.g., ICD-O-3 codes for malignant and benign tumors) and which demographic variables were included. Clarifying this would improve reproducibility and transparency.

Answer: We thank the reviewer for the comments.  We have revised this paragraph accordingly as follows (Lines 76-80):

The current analysis examined the time trends of incidence rates of overall benign (behavior classified as borderline and in situ) and malignant (behavior as malignant) brain cancer (including ICD-O-3 site and histology codes: brain- C710-C719 (excluding 9050-9055, 9140, 9530-9539, 9590-9993), cranial nerves other nervous system- C710-C719 (9530-9539), C700-C709 as well as C720-C729 (excluding 9050-9055, 9590-9993)).

The results mention that the p-value for benign acoustic neuroma (p = 0.8237) was not significant, but the implication of this result could be better explained. Consider stating clearly: “The lack of significant change in the incidence of benign acoustic neuroma (p = 0.8237) suggests that mobile phone use is unlikely to be associated with this tumor type.”

Answer: We thank the reviewer for the comments.  We have revised accordingly in lines: 20-21 and in results (lines 128-130).

The manuscript contains several minor grammatical and typographical issues. For example, the phrase “that that” in the introduction (line 31) should be corrected. A thorough language and grammar check throughout the document is recommended.

Answer: We thank the reviewer for the comments.  We have revised accordingly.

The introduction should better articulate the rationale for this study by highlighting the gap it addresses—particularly the need for up-to-date, U.S.-based trend data given rising mobile phone usage. The ecological design and absence of individual-level exposure data should also be acknowledged upfront.

Answer: Almost all studies of cell phones and cancer have been case-control designs, and to a lesser extent cohort studies. Another methodology that can be used to address this question is the analysis of trends, where rising incidence or mortality indicates a potentially new risk factor. This might be especially useful to account for a long hypothetical latency (lines 60-64).

The discussion section should be expanded to further engage with broader literature, including large-scale cohort studies and meta-analyses. This will place the current findings in better context and demonstrate their relevance within the wider field.

 Answer: We thank the reviewer for the comments.  We have revised accordingly in discussion (Lines 176-186).

A systematic review of 63 studies of radiofrequency fields on cancer risk commissioned by the World Health Organization concluded with moderate certainty a lack of risk from mobile phone use [50]. Inherent in these studies are the usual concerns in study designs: exposure assessment validity, recall bias, latency issues. Among these, the recent Cohort Study on Mobile Phones and Health (COSMOS) is a multi-center study in Europe that had detailed self-reported phone usage as well as supplemental baseline data on phone usage from the network operator [22]. The study did not find associations of lifetime mobile phone use with glioma, acoustic neuroma, or meningioma. In the UK Million Women Study, there was also no association of mobile phone use with temporal lobe gliomas [24].

While the statistical analysis is sound, the manuscript should acknowledge potential confounding variables such as changes in environmental exposures or diagnostic practices over time. This discussion will enhance the validity of the interpretations.

 Answer: We thank the reviewer for the comments.  We have revised accordingly in the discussion (Lines 241-244).

Also, potential confounding variables such as changes in environmental exposures or diagnostic practices over time could not be adjusted for in the model due to lack of this information in SEER.

The graphs and tables are generally well-constructed but could be improved with clearer axis labels, larger font sizes, and more descriptive captions to aid reader comprehension. If possible, visual indicators of statistical significance should be added to key graphs.

 Answer: We thank the reviewer for the comments.  We have revised accordingly.

The observed increase in benign tumors—particularly meningiomas—warrants a more detailed explanation. The authors suggest improved imaging is responsible; however, citing evidence for this trend (e.g., increase in MRI usage) would strengthen this interpretation.

 Answer: We thank the reviewer for the comments.  We have revised accordingly as follows (Lines 157-159).

Since 2000, there has been an annual increase in the use of noninvasive medical imaging for accurately diagnosing and treating diseases/conditions of the head, which has increased incidental detection of asymptomatic meningiomas [46, 47].

The study design, being ecological, limits causal inference. This should be emphasized more clearly in both the discussion and the abstract. A brief acknowledgment of this limitation in the conclusion would also enhance transparency.

 Answer: We thank the reviewer for the comments.  We have revised accordingly as follows in the discussion (Lines 225-226).

Study design, being ecological, limits causal inference.

While the study assesses population-level trends, individual exposure levels are not captured. The proxy measure (mobile phone subscriptions) does not account for frequency, duration, or type of use, which are essential factors. The authors should emphasize this limitation and suggest the need for future studies incorporating more granular usage data.

Answer: We thank the reviewer for the comments.  We have revised accordingly as follows (Lines 226-230 in discussion).

While the study assesses population-level trends, individual exposure levels are not captured. The proxy measure (mobile phone subscriptions) does not account for frequency, duration, or type of use, which are essential factors that are not captured in the data. Future studies should incorporate more granular usage data.

The discussion on biological mechanisms is underdeveloped. The authors should include either relevant experimental evidence or propose potential mechanisms through which RF-EMF exposure might theoretically affect brain tissue. Alternatively, explain more clearly why no mechanism has been established despite widespread exposure.

 Answer: We thank the reviewer for the comments. We have added a discussion (lines 196-201) that it is widely established that there is no known mechanism by which radiofrequency fields can cause cancer (https://www.cancer.gov/about-cancer/causes-prevention/risk/radiation/electromagnetic-fields-fact-sheet). It does not damage DNA. Studies have attempted to find possible indirect mechanisms, such as changes in melatonin levels or oxidative stress, but there is little convincing evidence. Most animal studies have not shown any relationship with tumor incidence.

Given the importance of biological plausibility in interpreting epidemiologic trends, the manuscript would be significantly improved by either conducting or referencing experimental studies. This could include: In vitro or in vivo studies examining RF-EMF effects on neural tissues. SAR distribution simulations in adult and pediatric head models. Studies investigating oxidative stress, DNA damage, or apoptosis due to RF exposure. If direct experiments are not possible, referencing high-quality experimental literature would help bridge the gap between epidemiological and biological evidence.

 Answer: We thank the reviewer for the comments.  We have revised accordingly in lines 208-210.

The lack of increase in temporal lobe or acoustic neuroma tumors—areas most exposed to RF radiation suggests that mobile phones may not contribute to the brain cancer.

The issue of latency should be discussed more extensively. Although the 20-year study window is substantial, it remains possible that some cancers, particularly in younger users, might manifest beyond this period. More detailed age-stratified analyses would be useful.

Answer: We thank the reviewer for the comments.  We have included children and adolescents in the sub analysis.

The authors should provide clearer interpretations of non-significant findings. For example, the lack of increase in temporal lobe or acoustic neuroma tumors—areas most exposed to RF radiation—should be emphasized as important negative evidence.

Answer: We thank the reviewer for the comments.  We have revised accordingly as follows (Lines 208-210).

The lack of increase in temporal lobe or acoustic neuroma tumors—areas most exposed to RF radiation suggests that mobile phones may not contribute to the brain cancer.

Statistical model assumptions (e.g., linearity in regression trends) should be briefly described in the methods. Clarifying why SEER*Stat was appropriate and what assumptions were made would improve methodological transparency.

Answer: We thank the reviewer for the comments.  We have revised accordingly as follows in Lines 102-107.

Assuming the linearity in regression trends holds, by fitting linear regression models with each calendar year at diagnosis (2000-2021) as an independent variable, changes (significant increasing/decreasing or stable) in time trends of age-adjusted incidence rates were assessed using the annual percentage change (APC) based on least square methods and corresponding intervals (CIs) estimated by SEER*Stat 8.4.3, as recommended by SEER website.

The conclusions should include more forward-looking suggestions. For example, recommend longitudinal cohort studies with operator-recorded usage data or advanced imaging biomarkers for early detection.

Answer: We thank the reviewer for the comments.  We have revised accordingly in conclusion (Lines 252-254).

Longitudinal cohort studies with operator-recorded usage data or advanced imaging biomarkers for early detection are needed in the near future to confirm the findings.

Round 2

Reviewer 1 Report

Comments and Suggestions for Authors

The revisions made are acceptable. I have no further suggestion for improvement. 

Reviewer 2 Report

Comments and Suggestions for Authors

The authors have addressed all of the reviewers comments. The manuscript is suitable for acceptance in its current form.

Reviewer 3 Report

Comments and Suggestions for Authors

The authors have thoroughly addressed all reviewer comments, including clarifications on statistical methodology, limitations of the ecological study design, biological plausibility, and interpretation of both significant and non-significant findings. They have also improved the manuscript’s clarity, organization, and contextualization within the literature.